# GLOBALLY INJECTIVE RELU NETWORKS

## ABSTRACT

Injectivity plays an important role in generative models where it enables inference; in inverse problems and compressed sensing with generative priors it is a precursor to well posedness. We establish sharp characterizations of injectivity of fully-connected and convolutional ReLU layers and networks. First, through a layerwise analysis, we show that an expansivity factor of two is necessary and sufficient for injectivity by constructing appropriate weight matrices. We show that global injectivity with iid Gaussian matrices, a commonly used tractable model, requires larger expansivity between 3.4 and 10.5. We also characterize the stability of inverting an injective network via worst-case Lipschitz constants of the inverse. We then use arguments from differential topology to study injectivity of deep networks and prove that any Lipschitz map can be approximated by an injective ReLU network. Finally, using an argument based on random projections, we show that an end-to-end—rather than layerwise—doubling of the dimension suffices for injectivity. Our results establish a theoretical basis for the study of nonlinear inverse and inference problems using neural networks.

## 1 INTRODUCTION

Many applications of deep neural networks require inverting them on their range. Given a neural network $N : \mathcal{Z} \to \mathcal{X}$, where $\mathcal{X}$ is often the Euclidean space $\mathbb{R}^m$ and $\mathcal{Z}$ is a lower-dimensional space, the map $N^{-1} : N(\mathcal{Z}) \to \mathcal{Z}$ is only well-defined when $N$ is injective. The issue of injectivity is particularly salient in two applications: generative models and (nonlinear) inverse problems.

Generative networks model a complicated distribution $p_X$ over $\mathcal{X}$ as a pushforward of a simple distribution $p_Z$ through $N$. Given an $x$ in the range of $N$, inference requires computing $p_Z(N^{-1}(x))$ which is well-posed only when $N$ is injective. In the analysis of inverse problems (Arridge et al., 2019), uniqueness of a solution is a key concern; it is tantamount to injectivity of the forward operator. Given a forward model that is known to yield uniqueness, a natural question is whether we can design a neural network that approximates it arbitrarily well while preserving uniqueness. Similarly, in compressed sensing with a generative prior $N$ and a possibly nonlinear forward operator $A$ injective on the range of $N$, we seek a latent code $z$ such that $A(N(z))$ is close to some measured $y = A(x)$. This is again only well-posed when $N$ can be inverted on its range (Balestriero et al., 2020). Beyond these motivations, injectivity is a fundamental mathematical property with numerous implications. We mention a notable example: certain injective generators can be trained with sample complexity that is polynomial in the image dimension (Bai et al., 2018).

### 1.1 OUR RESULTS

In this paper we study injectivity of neural networks with ReLU activations. Our contributions can be divided into layerwise results and multilayer results.

**Layerwise results.** For a ReLU layer $f : \mathbb{R}^n \to \mathbb{R}^m$ we derive sufficient and necessary conditions for invertibility on the range. For the first time, we construct deterministic injective ReLU layers with minimal expansivity $m = 2n$. We then derive specialized results for convolutional layers which are given in terms of filter kernels instead of weight matrices. We also prove upper and lower bounds on minimal expansivity of globally injective layers with iid Gaussian weights. This generalizes certain existing pointwise results (Theorem 2 and Appendix A.2). We finally derive the worst-case inverse

Lipschitz constant for an injective ReLU layer which yields stability estimates in applications to inverse problems.

**Multilayer results.** A natural question is whether injective models are sufficiently expressive. Using techniques from differential topology we prove that injective networks are *universal* in the following sense: if a neural network $N_1 : \mathcal{Z} \to \mathbb{R}^{2n+1}$ models the data, $\mathcal{Z} \subset \mathbb{R}^n$, then we can approximate $N_1$ by an injective neural network $N_2 : \mathcal{Z} \to \mathbb{R}^{2n+1}$. As $N_2$ is injective, the image set $N_2(\mathcal{Z})$ is a Lipschitz manifold. We then use an argument based on random projections to show that an *end-to-end* expansivity by a factor of $\approx 2$ is enough for injectivity in ReLU networks, as opposed to layerwise 2-expansivity implied by the layerwise analysis.

We conclude with preliminary numerical experiments to show that imposing injectivity improves inference in GANs while preserving expressivity.

## 1.2 WHY GLOBAL INJECTIVITY?

The attribute "global" relates to global injectivity of the map $N : \mathcal{Z} \to \mathbb{R}^m$ on the low-dimensional latent space $\mathcal{Z}$, but it does not imply global invertibility over $\mathbb{R}^m$, only on the range $N(\mathcal{Z}) \subset \mathbb{R}^m$. If we train a GAN generator to map iid normal latent vectors to real images from a given distribution, we expect that any sampled latent vector generates a plausible image. We thus desire that any $N(z)$ be produced by a unique latent code $z \in \mathcal{Z}$. This is equivalent to global injectivity, or invertibility on the range. Our results relate to the growing literature on using neural generative models for compressed sensing (Bora et al., 2017). They parallel the related guarantees for sparse recovery where the role of the low-dimensional latent space is played by the set of all $k$-sparse vectors. One then looks for matrices which map *all* $k$-sparse vectors to distinct measurements (Foucart & Rauhut, 2013). As an example, in the illustration in Figure 1 images coresponding to latent codes in orange, brown, and violet wedges cannot be compressively sensed. Finally, a neural network is often trained to directly reconstruct an image $x$ from its (compressive) low-dimensional measurements $y = A(x)$ without introducing any generative models. In this case, whenever $A$ is Lipschitz, it is immediate that the learned inverse must be injective.

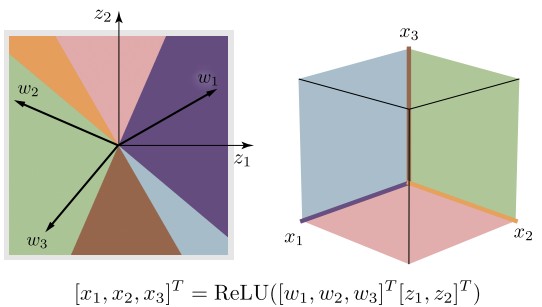

$$[x_1, x_2, x_3]^T = \mathrm{ReLU}([w_1, w_2, w_3]^T [z_1, z_2]^T)$$

Figure 1: An illustration of a ReLU layer $N :$ $\mathbb{R}^2 \to \mathbb{R}^3$, $x = N(z)$, that is not globally injective. Differently colored regions in the $z$-space are mapped to regions of the same color in the $x$-space. While $N$ is locally injective in the pink, blue and green wedges in $z$-space, the orange, brown, and violet wedges are mapped to coordinate axes. $N$ is thus *not* injective on these wedges. This prevents construction of an inverse in the range of $N$.

## 1.3 RELATED WORK

Closest to our work are the papers of Bruna et al. (2013), Hand et al. (2018) and Lei et al. (2019). Bruna et al. (2013) study injectivity of pooling motivated by the problem of signal recovery from feature representations. They focus on $\ell^p$ pooling layers; their Proposition 2.2 gives a criterion similar to the DSS (Definition 1) and bi-Lipschitz bounds for a ReLU layer (similar to our Theorem 3). Unlike Theorems 1 and 3, their criterion and Lipschitz bound are in some cases not precisely aligned with injectivity; see Appendix E.1.

Compressed sensing with GAN priors requires inverting the generator on its range (Bora et al., 2017; Shah & Hegde, 2018; Wu et al., 2019; Mardani et al., 2018; Hand et al., 2018). Lei et al. (2019) replace the end-to-end inversion by the faster and more accurate layerwise inversion when each layer is injective. They show that with high probability a ReLU layer with an iid normal weight matrix can be inverted about a fixed point if the layer expands at least by a factor of 2.1. This result is related to our Theorem 2 which gives conditions for global injectivity or layers with random matrices. Hand & Voroninski (2017) show that when the weights of a ReLU network obey a certain weighted distribution condition, the loss function for the inversion has a strict descent direction everywhere

except in a small set. The condition is in particular satisfied by random matrices with expansivity $n_j = \Omega(n_{j-1} \log n_{j-1})$, where $n_j$ is the output dimension of layer $j$.

A continuous analogy of our convolutional construction (Definition 3) was considered by Mallat et al. (2018). They show that $\mathrm{ReLU}$ acts as a phase filter and that the layer is bi-Lipschitz and hence injective when the filters have a diverse phase and form a frame. Discretizing their model gives a statement related to Corollary 2 and Theorem 4.

Injectivity is automatic in invertible neural networks such as normalizing flows (Grover et al., 2018; Kingma & Dhariwal, 2018; Grathwohl et al., 2018). Specialized architectures with simple Jacobians give easy access to the likelihood (Dinh et al., 2014; 2016; Gomez et al., 2017), which facilitates application to inverse problems (Ardizzone et al., 2018). Inference with GANs can be achieved by jointly training a generative network and its inverse (Donahue et al., 2016; Dumoulin et al., 2016), which is well-defined when the generator is injective. Relaxed injective probability flows resemble GAN generators but are trained via approximate maximum likelihood (Kumar et al., 2020). Injectivity is promoted by keeping the smallest singular value of the Jacobian away from zero at the training examples, a necessary but not sufficient condition. In general, Jacobian conditioning improves GAN performance (Heusel et al., 2017; Odena et al., 2018). Finally, lack of injectivity interferes with disentanglement (Chen et al., 2016; Lin et al., 2019). In this context, injectivity seems to be a natural heuristic to increase latent space capacity without increasing its dimension (Brock et al., 2016).

**Notation.** Given a matrix $W \in \mathbb{R}^{m \times n}$ we define the notation $w \in W$ to mean that $w \in \mathbb{R}^n$ is a row vector of $W$. For a matrix $W \in \mathbb{R}^{m \times n}$ with rows $\{w_j\}_{j=1}^m$ and $x \in \mathbb{R}^n$, we write

$$S(x, W) := \{j \in [[m]]: \langle w_j, x \rangle \geq 0\} \tag{1}$$

and $S^c(x, W)$ for its complement, with $[[m]] = \{1, \ldots, m\}$. We let $\mathcal{NN}(n, m, L, \boldsymbol{m})$ be the family of functions $N_\theta : \mathbb{R}^n \to \mathbb{R}^m$ of the form

$$N(z) = W_{L+1}\phi_L(W_L \cdots \phi_2(W_2\phi_1(W_1 z + b_1) + b_2) \cdots + b_L) \tag{2}$$

Indices $\ell = 1, \ldots, L$ index the network layers, $b_\ell \in \mathbb{R}^{n_{\ell+1}}$ are the bias vectors, $W_\ell \in \mathbb{R}^{n_{\ell+1} \times n_\ell}$ are the weight matrices with $n_1 = n$, $n_L = m$, and $\phi_\ell$ are the nonlinear activation functions. We will denote $\mathrm{ReLU}(x) = \max(x, 0)$. We write $\boldsymbol{n} = (n_1, n_2, \ldots n_{L-1})$ and $\theta = (W_1, b_1, \ldots, W_L, b_L)$ for the parameters that determine the function $N_\theta$. We also write $\mathcal{NN}(n, m, L) = \bigcup_{\boldsymbol{n} \in \mathbb{Z}^{L-1}} \mathcal{NN}(n, m, L, \boldsymbol{n})$ and $\mathcal{NN}(n, m) = \bigcup_{L \in \mathbb{Z}} \mathcal{NN}(n, m, L)$.

## 2 Layerwise Injectivity of ReLU Networks

For a one-to-one activation function such as a leaky $\mathrm{ReLU}$ or a $\tanh$, it is easy to see that injectivity of $x \mapsto W_i x$ implies the injectivity of the layer. We therefore focus on non-injective $\mathrm{ReLU}$ activations.

### 2.1 Directed Spanning Set

Unlike in the case of a one-to-one activation, when $\phi(x) = \mathrm{ReLU}(x)$, $x \mapsto \phi(Wx)$ cannot be injective for all $x \in \mathbb{R}^n$ if $W$ is a square matrix. Noninjectivity of $\mathrm{ReLU}(Wx)$ occurs in some simple situations, for example if $Wx$ is not full rank or if $Wx \leq 0$ element-wise for a nonzero $x$, but these two conditions on their own are not sufficient to characterize injectivity. Figure 2 gives an example of a weight matrix that yields a noninjective layer that passes these two tests. In order to facilitate the full characterization of injectivity of $\mathrm{ReLU}$ layers, we define a useful theoretical device:

**Definition 1** (Directed Spanning Set)**.** Let $W \in \mathbb{R}^{n \times m}$. We say that $W$ has a directed spanning set (DSS) of $\Omega \subset \mathbb{R}^n$ with respect to a vector $x \in \mathbb{R}^n$ if there exists a $\hat{W}_x$ such that each row vector of $\hat{W}_x$ is a row vector of $W$,

$$\langle x, w_i \rangle \geq 0 \quad \text{for all } w_i \in \hat{W}_x, \tag{3}$$

and $\Omega \subset \mathrm{span}(\hat{W}_x)$. When omitted $\Omega$ is understood to be $\mathbb{R}^n$.

For a particular $x \in \mathbb{R}^n$, it is not hard to verify if a given $W = \{w_i\}_{i=1,\ldots,m}$ has a DSS of $\Omega$. One can simply let $\hat{W}_x = \{w_i \in \mathbb{R}^n : \langle x, w_i \rangle \geq 0\}$. Then, $W$ has a DSS for $\Omega$ with respect to $x$ if and only if $\hat{W}_x$ is a basis of $\Omega$, which can be checked efficiently. Note that having full rank is necessary for it

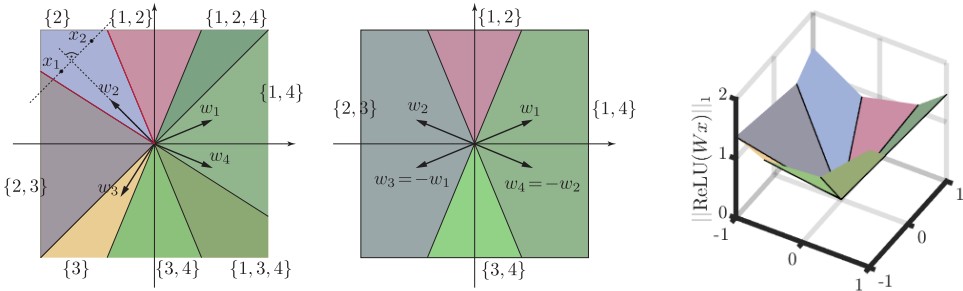

Figure 2: Illustration of the DSS definition. *Left:* A configuration of 4 vectors in $\mathbb{R}^2$ that do not have a DSS w.r.t. all $x \in \mathbb{R}^2$. In this case, the vectors do not generate an injective layer. The set of labels indicate which $w_j$ have positive inner product with vectors in the wedge; there are two wedges with only one such $\{w_j\}$; we have $\text{ReLU}(Wx_1) = \text{ReLU}(Wx_2)$. *Center:* A configuration where four vectors have a DSS for all $x \in \mathbb{R}^2$. These vectors correspond to a minimally-expansive injective layer; see Corollary 2. *Right:* A plot of $\|\text{ReLU}(Wx)\|_1$ where $W$ is given as in the left figure. Note that $x \mapsto \text{ReLU}(Wx)$ is linear in every wedge.

to have a DSS for *any* $x \in \mathbb{R}^n$. It is however not sufficient. For example, $\text{Id}_n \in \mathbb{R}^{n \times n}$, the identity matrix in $\mathbb{R}^n$, is clearly full rank, but it doesn't have a DSS of $\mathbb{R}^n$ w.r.t. $x = (-1, 0, \ldots, 0) \in \mathbb{R}^n$.

To check whether $W$ has a DSS for all $x \in \mathbb{R}^n$, note that $W$ partitions $\mathbb{R}^n$ into open wedges $S_k$, $\mathbb{R}^n = \bigcup_k S_k$, with constant sign patterns. That is, for $x_1, x_2 \in S_k$, $\text{sign}(Wx_1) = \text{sign}(Wx_2)$. (See also proof of Theorem 2 in Appendix A.2.2.) Checking whether $W$ has a DSS for all $x$ is equivalent to checking that for every wedge there are at least $n$ vectors $W_k \subset W$ with positive sign, $\langle w, x \rangle > 0$ for $x \in S_k$, and that $W_k$ spans $\mathbb{R}^n$. Since the number of wedges can be exponential in $m$ and $n$ (Winder, 1966) this suggests an exponential time algorithm. We also note the connection between DSS and spark in compressed sensing (Foucart & Rauhut, 2013), defined as the size of the smallest set of linearly dependent vectors in $W$. If every wedge has $n$ or more positive signs, then full spark $n + 1$ is sufficient for $W$ to have a DSS w.r.t all $x \in \mathbb{R}^n$. Computing spark is known to be NP-hard (Tillmann & Pfetsch, 2013); whether one can do better for DSS remains an open question.

## 2.2 FULLY CONNECTED LAYER

The notion of a DSS immediately leads to our main result for fully connected layers.

**Theorem 1** (Conditions for Injectivity of $\text{ReLU}(Wx)$). *Let* $W \in \mathbb{R}^{m \times n}$ *where* $n > 1$ *be a matrix with row vectors* $\{w_j\}_{j=1}^m$*, and* $\text{ReLU}(y) = \max(y, 0)$*. The function* $\text{ReLU}(W(\cdot)) \colon \mathbb{R}^n \to \mathbb{R}^m$ *is injective if and only if* $W$ *has a DSS w.r.t every* $x \in \mathbb{R}^n$*.*

The question of injectivity in the case when $b = 0$ and when $b \neq 0$ are very similar and, as Lemma 1 shows, the latter question is equivalent to the former on a restricted weight matrix.

**Lemma 1** (Injectivity of $\text{ReLU}(Wx+b)$). *Let* $W \in \mathbb{R}^{m \times n}$ *and* $b \in \mathbb{R}^m$*. The function* $\text{ReLU}(W(\cdot) + b) \colon \mathbb{R}^n \to \mathbb{R}^m$ *is injective if and only if* $\text{ReLU}(W|_{b \geq 0} \cdot)$ *is injective, where* $W|_{b \geq 0} \in \mathbb{R}^{m \times n}$ *is row-wise the same as* $W$ *where* $b_i \geq 0$*, and is a row of zeroes when* $b_i < 0$*.*

*Remark* 1 (Injectivity of $\text{ReLU}(Wx)$, Positive $x$). Without normalization strategies, inputs of all but the first layer are element-wise non-negative. One can ask whether $\text{ReLU}(Wx)$ is injective when $x$ is restricted to be element-wise non-negative. Following the same argument as in Theorem 1, we find that $W$ must have a DSS w.r.t. $x \in \mathbb{R}^n$ for every $x$ that is element-wise non-negative. With normalizations strategies however (for example batch renormalization (Ioffe, 2017)) the full power of Theorem 1 and Lemma 1 may be necessary. In Appendix B we show that common normalization strategies such as batch, layer, or group normalization do not interfere with injectivity.

*Remark* 2 (Global vs Restricted Injectivity). Even when the conditions of Theorem 1 and Lemma 1 are not satisfied the network might be injective in some $\mathcal{X} \subset \mathbb{R}^n$. Indeed, $N(x) = \text{ReLU}(\text{Id}_n \cdot)$ is in general not injective, but it is injective in $\mathcal{X} = \{x \in \mathbb{R}^n \colon x_i > 0 \text{ for } i = 1, \ldots, n\}$, a convex and open set. Theorem 1 and Lemma 1 are the precise criteria for single-layer injectivity for all $x \in \mathbb{R}^n$.

The above layerwise results imply a sufficient condition for injectivity of deep neural networks.

**Corollary 1** (layerwise Injectivity Implies End-to-End Injectivity). *Let $N\colon \mathbb{R}^n \to \mathbb{R}^m$ be a deep neural network of the form (2). If each layer $\phi_\ell(W_\ell \cdot + b_\ell)\colon \mathbb{R}^{n_i} \to \mathbb{R}^{n_{i+1}}$ is injective, then so is $N$.*

Note that Corollary 1 is sufficient but not necessary for injectivity of the entire network. Consider for example an injective layer $\mathrm{ReLU}(W)$, and $N(x) = \mathrm{ReLU}(\mathrm{Id}_m \cdot \mathrm{ReLU}(W(x)))$, where $I$ is the identity matrix. Clearly,

$$\mathrm{ReLU}(\mathrm{Id}_m \cdot \mathrm{ReLU}(W(x))) = \mathrm{ReLU}(\mathrm{ReLU}(W(x))) = \mathrm{ReLU}(W(x)) \tag{4}$$

so $N$ is injective, but it fails the criterion of Corollary 1 (at the $\mathrm{ReLU}(I\cdot)$ layer).

An important implication of Theorem 1 is the following result on minimal expansivity.

**Corollary 2** (Minimal Expansivity). *For any $W \in \mathbb{R}^{m\times n}$, $\mathrm{ReLU}(W\cdot)$ is non-injective if $m < 2 \cdot n$. If $W \in \mathbb{R}^{2n\times n}$ satisfies Theorem 1 and Lemma 1, than up to row rearrangement $W$ can be written as*

$$W = \begin{bmatrix} B \\ -DB \end{bmatrix} \tag{5}$$

*where $B, D \in \mathbb{R}^{n\times n}$ and $B$ is a basis and $D$ a diagonal matrix with strictly positive diagonal entries.*

While $m < 2n$ immediately precludes injectivity, Corollary 2 gives a simple recipe for the construction of minimally-expansive injective layers with $m = 2n$. To the best of our knowledge this is the first such result. Further, we can also use Corollary 2 to build weight matrices for which $m > 2n$. For example, the matrix

$$W = \begin{bmatrix} B \\ -DB \\ M \end{bmatrix},$$

where $B \in \mathbb{R}^{n\times n}$ has full rank, $D \in \mathbb{R}^{n\times n}$ is a positive diagonal matrix, and $M \in \mathbb{R}^{m-2n\times n}$ is arbitrary, has a DSS w.r.t. every $x \in \mathbb{R}^n$. This method can be used in practice to generate injective layers by using standard spectral regularizers to ensure that $B$ is a basis (Cisse et al., 2017). In Section 3 we will show that, in fact, an *end-to-end* rather than layerwise doubling of dimension is sufficient for injectivity.

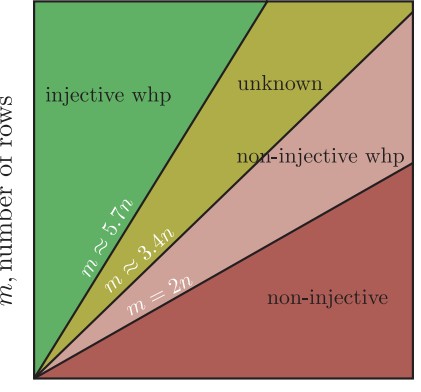

Figure 3: A visualization of regions where asymptotics of $\mathcal{I}(m,n)$ in Theorem 2 are valid.

Previous work that sought injectivity uses random-weight models. Lei et al. (2019) show that a layer is invertible *about a point* in the range provided that $m \geq 2.1n$ and $W$ is iid Gaussian. In Appendix A.3 we show that $m = 2.1n$ is not enought to guarantee injectivity. In fact, we show that with high probability an iid Gaussian weight matrix $W$ yields a *globally* invertible ReLU layer, when $W$ is sufficiently expansive, and conversely does not satisfy Theorem 1 if it is not expansive by at least a factor of 3.4:

**Theorem 2** (Injectivity for Gaussian Weights). *Let $\mathcal{I}(m,n) = \mathbb{P}\{x \mapsto \mathrm{ReLU}(Wx) \text{ is injective}\}$ with the entries of $W$ iid standard normal and $c = m/n$ fixed. Then as $n \to \infty$,*

$$\mathcal{I}(m,n) \to 1 \text{ if } c \gtrapprox 10.5 \text{ and } \mathcal{I}(m,n) \to 0 \text{ if } c \lessapprox 3.4.$$

The parameter regions defined in Theorem 2 are illustrated in Figure 3.

We note that the study of neural networks with random weights is an important recent research direction (Pennington & Worah, 2017; Benigni & Péché, 2019). Pennington & Worah (2017) articulate this clearly, arguing that large complex systems, of which deep networks are certainly an example, can be profitably studied by approximating their constituent parts as random variables.

## 2.3 INVERSE LIPSCHITZ CONSTANT FOR NETWORKS

Because $\|\mathrm{ReLU}(Wx) - \mathrm{ReLU}(Wy)\| \leq \|W\| \|x - y\|$, it is clear that $\mathrm{ReLU}(W\cdot)$ is Lipschitz with constant $\|W\|$; whether the inverse is Lipschitz, and if so, with what Lipschitz constant, is less obvious. We can prove the following result (see Appendix C.1):

**Theorem 3** (Global Inverse Lipschitz Constant). *Let $W \in \mathbb{R}^{m \times n}$ have a DSS w.r.t. every $x \in \mathbb{R}^n$. There exists a $C(W) > 0$ such that for any $x_0, x_1 \in \mathbb{R}^n$,*

$$\|\text{ReLU}(Wx_0) - \text{ReLU}(Wx_1)\|_2 \geq C(W) \|x_0 - x_1\|_2 \tag{6}$$

*where $C(W) = \frac{1}{\sqrt{2m}} \min_{x \in \mathbb{R}^n} \sigma(W|_{S(x,W)})$, and $\sigma$ denotes the smallest singular value.*

This result immediately yields stability estimates when solving inverse problems by inverting (injective) generative networks. We note that the $(2m)^{-1/2}$ factor is essential; the naive "smallest singular value" estimate is too optimistic.

## 2.4 CONVOLUTIONAL LAYER

Since convolution is a linear operator we could simply apply Theorem 1 and Lemma 1 to the convolution matrix. It turns out, however, that there exists a result specialized to convolution that is much simpler to verify. In this section, capital letters such as $I, J$ and $K$ denote multi-indexes; the same holds for their element-wise limits such as $N$ and $M$. $N_j$ refers to the $j$th index in the multi-index $N$. We use the shortcuts $c \in \mathbb{R}^N$ for $c \in \mathbb{R}^{N_1} \times \cdots \times \mathbb{R}^{N_p}$, and $\sum_{I=1}^N c_I = \sum_{I_1=1}^{N_1} \cdots \sum_{I_p=1}^{N_p} c_{I_1,\ldots,I_p}$, for $I = (I_1, \ldots, I_p)$. The symbol 1 can refer to number 1 or a $p$-tuple $(1, \ldots, 1)$. Further, the notation $I \leq J$ means that $I_k \leq J_k$ for all $k = 1, \ldots, p$ and likewise for $\geq, <$ and $>$. If $I \not\leq J$, then there is at least one $k$ such that $I_k > J_k$.

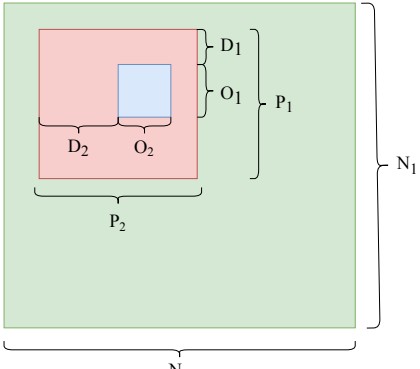

Figure 4: A visualization of the indices in (8) in two dimensions. The blue region is a kernel $c$ of width $O$, the pink region is the zero-padded box of width $P$, and $D$ is the offset of the kernel $c$ in the pink box. The entire signal is the green box of width $N$.

**Definition 2** (Convolution Operator). Let $c \in \mathbb{R}^O$. We say that $c$ is a convolution kernel of width $O$. Given $c$ and $x \in \mathbb{R}^N$, we define the convolution operator $C \in \mathbb{R}^{N \times N}$ with stride 1 as,

$$(Cx)_J = \sum_{I=1}^{O} c_{O-I+1} x_{J+I} = \sum_{I'=1+J}^{O+J} c_{O+J-I'+1} x_{I'}. \tag{7}$$

When $1 \not\leq K$ or $K \not\leq O$ we will set $c_K = 0$. We do not specify the boundary condition on $x$ (zero-padded, periodic, or otherwise), as our results hold generally.

**Definition 3** (Convolutional Layer). We say that a matrix $W \in \mathbb{R}^{M \times N}$ is a convolution operator, with $n_Q$ convolutions, provided that $W$ can be written (up to row rearrangement) in the form $W = \begin{bmatrix} C_1^T & C_2^T & \cdots & C_{n_Q}^T \end{bmatrix}^T$ where each $C_k$ is a convolution operator for $k = 1, \ldots, n_Q$. A neural network layer for which $W$ is a convolution operator is called a convolution layer.

Definitions 2 and 3 automatically model the standard multi-channel convolution used in practice. For 2D images of size $512 \times 512$, $n_c$ input channels, and $3 \times 3$ convolutions, we simply let $x \in \mathbb{R}^N$ with $N = (512, 512, n_c)$ and $c \in \mathbb{R}^O$ with $O = (3, 3, n_c)$.

To state our main result for convolutions we also need to define the set of zero-padded kernels for $c \in \mathbb{R}^O$. Think of a multi-index as a box (or a hyperrectangle). Let $P$ be a multi-index such that $O$ "fits" in $P$. Then we define

$$\mathcal{Z}_P(c) = \{D \in \mathbb{R}^P \ : \ d \text{ is a shift of } c \text{ within the box } P\}. \tag{8}$$

**Theorem 4** (Sufficient Condition for Injectivity of Convolutional Layer). *Suppose that $W \in \mathbb{R}^{M \times N}$ is a convolution layer with convolutions $\{C_k\}_{k=1}^q$, and corresponding kernels $\{c_k\}_{k=1}^q$. If for any $P$,*

$$W|_{\mathcal{Z}_P} := \bigcup_{k=1}^q \mathcal{Z}_P(c_k) \tag{9}$$

*has a DSS for $\mathbb{R}^P$ with respect to all $x \in \mathbb{R}^P$, then $\text{ReLU}(W \cdot)$ satisfies Theorem 1.*

Theorem 4 applies to multi-channel convolution of width $(O, n_c) = (O_1, \ldots, O_p, n_c)$ provided that we choose a $(P, n_c)$ such that $O \leq P$. This theorem shows that if a convolution operator $W$ has a DSS w.r.t. vectors with support in the pink region in Figure 4, then it has a DSS w.r.t. vectors supported on the green region as well.

## 3 UNIVERSALITY AND EXPANSIVITY OF DEEP INJECTIVE NETWORKS

We now consider general properties of deep injective networks. Note that a neural network $N_\theta \in \mathcal{NN}(n, m)$ is Lipschitz smooth: there is $L_0 > 0$ such that $|N_\theta(x) - N_\theta(y)| \leq L_0 |x - y|$ for all $x, y \in \mathbb{R}^n$. If $N_\theta : \mathbb{R}^n \to \mathbb{R}^m$ is also injective, results of basic topology imply that for any bounded and closed set $B \subset \mathbb{R}^n$ the map $N_\theta : B \to N_\theta(B)$ has a continuous inverse $N_\theta^{-1} : N_\theta(B) \to B$ and the sets $B$ and its image $N_\theta(B)$ are homeomorphic. Thus, for example, if $Z$ is a random variable supported on the cube $[0, 1]^n \subset \mathbb{R}^n$, the sphere $\mathbb{S}^{n-1} \subset \mathbb{R}^n$, or the torus $\mathbb{T}^{n-1} \subset \mathbb{R}^n$, we see that $N_\theta(Z)$ is a random variable supported on a set in $\mathbb{R}^m$ that is homeomorphic to an $n$-dimensional cube or an $n - 1$ dimensional sphere or a torus, respectively. This means that injective neural networks can model random distributions on surfaces with prescribed topology. Moreover, if $N_\theta : \mathbb{R}^n \to \mathbb{R}^m$ is a decoder, all objects $x \in N_\theta(\mathbb{R}^n)$ correspond to the unique code $z$ such that $x = N_\theta(z)$.

A fundamental property of neural networks is that they can uniformly approximate any continuous function $f : \mathcal{Z} \to \mathbb{R}^m$ defined in a bounded set $\mathcal{Z} \subset \mathbb{R}^n$. For general dimensions $n$ and $m$, the injective neural networks do not have an analogous property. For instance, if $f : [-\pi, \pi] \to \mathbb{R}$ is the trigonometric function $f(x) = \sin(x)$, it is easy to see that there is no injective neural network (or any other injective function) $N_\theta : [-\pi, \pi] \to \mathbb{R}$ such that $|f(x) - N_\theta(x)| < 1$. Consider, however, the following trick: add two dimensions in the output vector and consider the map $F : [-\pi, \pi] \to \mathbb{R}^3$ given by $F(x) = (0, 0, \sin(x)) \in \mathbb{R}^3$. When $f$ is approximated by a one-dimensional, non-injective ReLU network $f_\theta : [-\pi, \pi] \to \mathbb{R}$ and $\alpha > 0$ is small, then the neural network $N_\theta(x) = (\alpha\phi(x), \alpha\phi(-x), f_\theta(x))$ is an injective map that approximates $F$.

In general, as ReLU networks are piecewise affine maps, it follows in the case $m = n$ that if a ReLU network $N_\theta : \mathbb{R}^n \to \mathbb{R}^n$ is injective, it has to be surjective (Scholtes, 1996, Thm. 2.1). This is a limitation of injective neural networks when $m = n$. Building on these ideas we show that when the dimension of the range space is sufficiently large, $m \geq 2n + 1$, injective neural networks become sufficiently expressive to universally model arbitrary continuous maps.

**Theorem 5** (Universal Approximation with Injective Neural Networks). *Let $f : \mathbb{R}^n \to \mathbb{R}^m$ be a continuous function, where $m \geq 2n + 1$, and $L \geq 1$. Then for any $\varepsilon > 0$ and compact subset $\mathcal{Z} \subset \mathbb{R}^n$ there exists a neural network $N_\theta \in \mathcal{NN}(n, m)$ of depth $L$ such that $N_\theta : \mathbb{R}^n \to \mathbb{R}^m$ is injective and*

$$|f(x) - N_\theta(x)| \leq \varepsilon, \quad \text{for all } x \in \mathcal{Z}. \tag{10}$$

Before describing the proof, we note that this result also holds for leaky ReLU networks with no modification. The key fact used in the proof is that the network is a piecewise affine function.

To prove this result, we combine the approximation results for neural networks (e.g., Pinkus's density result for shallow networks (Pinkus, 1999, Theorem 3.1) or Yarotsky's result for deep neural networks (Yarotsky, 2017)), with the Lipschitz-smooth version of the generic projector technique. This technique from differential topology is used for example to prove the easy version of the Whitney's embedding theorem (Hirsch, 2012, Chapter 2, Theorem 3.5). Related random projection techniques have been used earlier in machine learning and compressed sensing (Broomhead & Kirby, 2000; 2001; Baraniuk & Wakin, 2009; Hegde et al., 2008; Iwen & Maggioni, 2013).

The proof of Theorem 5 is based on applying the above classical results to locally approximate the function $f : \mathbb{R}^n \to \mathbb{R}^m$ by some (possibly non-injective) ReLU-based neural network $F_\theta : \mathbb{R}^n \to \mathbb{R}^m$, and augment it by adding additional variables, so that $H_\theta(x) = (x, F_\theta(x))$ is an injective map $H_\theta : \mathbb{R}^n \to \mathbb{R}^{n+m}$. The image of this map, $M = H_\theta(\mathbb{R}^n)$, is an $n$-dimensional, Lipschitz-smooth submanifold of $\mathbb{R}^{n+m}$. The dimension of $m + n$ is larger than $2n + 1$, which implies that for a randomly chosen projector $P_1$ that maps $\mathbb{R}^{n+m}$ to a subspace of dimension $m + n - 1$, the restriction of $P_1$ on the submanifold $M$, that is, $P_1 : M \to P_1(M)$, is injective. By applying $n$ random projectors, $P_1, \ldots, P_n$, we have that $N_\theta = P_n \circ P_{n-1} \circ \cdots \circ P_1 \circ H_\theta$ is an injective map whose image is in a $m$ dimensional linear space. These projectors can be multiplied together, but are

generated sequentially. By choosing the projectors $P_j$ in a suitable way, the obtained neural network $N_\theta$ approximates the map $f$.

We point out that in the proof of Theorem 5 it is crucial that we first approximate function $f$ by a neural network and then apply to it a generic projection to make the neural network injective. Indeed, doing this in the opposite order may fail as an arbitrarily small deformation (in $C(\mathbb{R}^n)$) of an injective map may produce a map that is non-injective.

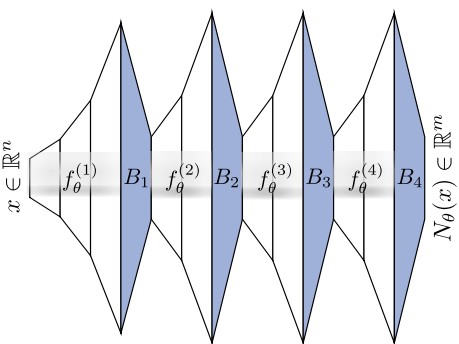

Figure 5: An illustration of an injective deep neural network that avoids expansivity as described by Corollary 3. White trapezoids are expansive weight matrices satisfying Theorem 1, and the blue trapezoids are random projectors that reduce dimension while preserving injectivity.

The proof of Theorem 5 implies the following corollary on cascaded neural networks where the dimensions of the hidden layers can both increase and decrease (see Figure 5):

**Corollary 3.** *Let $n, m, d_j \in \mathbb{Z}_+$, $j = 0, 1, \ldots, 2k$ be such that $d_0 = n$, $d_{2k} = m \geq 2n + 1$ and $d_j \geq 2n + 1$ for even indexes $j \geq 2$. Let*

$$F_k = B_k \circ f_\theta^{(k)} \circ B_{k-1} \circ f_\theta^{(k-1)} \circ \cdots \circ B_1 \circ f_\theta^{(1)}$$

*where $f_\theta^{(j)} : \mathbb{R}^{d_{2j-2}} \to \mathbb{R}^{d_{2j-1}}$ are injective neural networks and $B_j \in \mathbb{R}^{d_{2j-1}} \to \mathbb{R}^{d_{2j}}$ are random matrices whose joint distribution is absolutely continuous with respect to the Lebesgue measure of $\prod_{j=1}^{k}(\mathbb{R}^{d_{2j} \times d_{2j-1}})$. Then the neural network $F_k : \mathbb{R}^n \to \mathbb{R}^m$ is injective almost surely.*

Observe that in Corollary 3 the weight matrices $B_j$ and $B_{j'}$ may not be independent. Moreover, the distribution of the matrices $B_j$ may be supported in an arbitrarily small set, that is, a small random perturbation of the weight matrices make the function $F_k$ injective. Corollary 3 characterizes deep globally injective networks, while avoiding the exponential growth in dimension as indicated by Corollary 2.

## 4 NUMERICAL EXPERIMENT: INJECTIVE GANS IMPROVE INFERENCE

We devise a simple experiment to demonstrate that 1) one can construct injective networks using Corollary 2 that 2) perform as well as the corresponding non-injective networks, while 3) being better at inference problems. One way to do inference with GANs is to train a so-called inference network $I_{\text{net}} : \mathcal{X} \to \mathcal{Z}$ jointly with the generator $G : \mathcal{Z} \to \mathcal{X}$ so that $I(G(z)) \approx z$ (Donahue et al., 2016; Dumoulin et al., 2016); $I_{\text{net}}$ is trained to invert $G$ on its range. This is used to evaluate the likelihood of an image $x$ in the range of $G$ as $p_Z(I_{\text{net}}(x))$. However, for $p_Z(I_{\text{net}}(G(z)) = p_Z(z)$ to hold, $G$ must be injective—otherwise the distribution of $I_{\text{net}}(G(z))$ will be different from that of $z$. One would thus hope that the distribution of $I_{\text{net}}(G(z))$ will be closer to that of $z$ if we enforce injectivity. If that is the case and the sample quality is comparable, then we have a practical tool to improve inference.

We use the DCGAN (Radford et al., 2015) architecture with the same hyperparameters, layer sizes and normalization strategies for the regular GAN and the injective GAN; see Appendix F. Note that the original DCGAN uses several ReLU conv layers with an expansivity of 2 per layer. By Corollary 2 these layers are non-injective. Therefore, the filters of the DCGAN are modified to be of the form $[C; -s^2 C]$ in order to build an injective GAN. We train the networks to generate $64 \times 64 \times 3$ images and draw the latent code $z \in \mathbb{R}^{256}$ iid from a standard normal distribution. We test on CelebA (Liu et al., 2015) and FFHQ (Karras et al., 2019) datasets. To get a performance metric, we fit Gaussian distributions $\mathcal{N}(\mu, \Sigma)$ and $\mathcal{N}(\mu_{\text{inj}}, \Sigma_{\text{inj}})$ to $G(z)$ and $G_{\text{inj}}(z)$. We then compute the Wasserstein-2 distance $\mathcal{W}_2$ between the distribution of $z \sim \mathcal{N}(0, I_{256})$ and the two fitted Gaussians using the closed-form expression for $\mathcal{W}_2$ between Gaussians, $\mathcal{W}_2^2(\mathcal{N}(\mu_1, \Sigma_1), \mathcal{N}(\mu_2, \Sigma_2)) = \|\mu_1 - \mu_2\|^2 + \text{Tr}(\Sigma_1 + \Sigma_2 - 2(\Sigma_1 \Sigma_2)^{1/2})$. We summarize the results in Table 1. Despite the restrictions on the weights of injective generators, their performance on popular GAN metrics—Fréchet inception distance (FID) (Heusel et al., 2017) and inception score (IS) (Salimans et al., 2016)—is comparable to the standard GAN while inference improves. That the generated samples are indeed comparable to the standard GAN can be also gleaned from the from the figure in Appendix F.

Table 1: Injectivity improves inference without sacrificing performance.

| Dataset | Type of $G$ | Inception Score↑ | Fréchet Inception Distance↓ | $\mathcal{W}_2^2(\mathbb{P}_{\hat{z}}, \mathbb{P}_z)$↓ |
|---------|-------------|------------------|------------------------------|------------------------------------------------------------|
| *CelebA* | Injective | $2.24 \pm 0.09$ | $39.33 \pm 0.41$ | **18.59** |
|          | Regular   | $2.22 \pm 0.16$ | $50.56 \pm 0.52$ | 33.85 |
| *FFHQ*   | Injective | $2.56 \pm 0.15$ | $61.22 \pm 0.51$ | **9.87** |
|          | Regular   | $2.57 \pm 0.16$ | $47.23 \pm 0.90$ | 19.63 |

## 5 CONCLUSION

We derived and explored conditions for injectivity of ReLU neural networks. In contrast to prior work which looks at random weight matrices, our characterizations are deterministic and derived from first principles. They are also sharp in that they give sufficient and necessary conditions for layerwise injectivity. Our results apply to any network of the form (2)—they only involve weight matrices but make no assumptions about the architecture. We included explicit constructions for minimally expansive networks that are injective; interestingly, this simple criterion already improves inference in our preliminary experiments. The results on universality of injective neural networks further justify their use in applications; they also implicitly justify the various Jacobian conditioning strategies when learning to generate real-world data. Further, injective neural networks are topology-preserving homeomorphisms which opens applications in computational topology and establishes connections to tools such as self-organizing maps. Analysis of deep neural networks has an analogue in the analysis of inverse problems where one studies uniqueness, stability and reconstruction. Uniqueness coincides with injectivity, quantitative stability with the Lipschitz constant of the inverse, and, following Lei et al. (2019), from a linear program we get a reconstruction in the range.

## ACKNOWLEDGMENTS

We are grateful to Daniel Paluka and Charles Clum for pointing out a problem with the bound on the sum of binomial coefficients used to prove Theorem 2.

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
