# OpenReview forum: "Globally Injective ReLU networks"
_ICLR.cc/2021/Conference — Reject_

### Official Review · AnonReviewer4 · 2020-10-28
**The paper studies some mathematically sound problems but not necessarily useful for the ML community**

**Rating:** 5
**Confidence:** 4

**Review:**

This paper studies some conditions when a ReLU network is injective. For a single ReLU layer, it derives sufficient and necessary conditions for a deterministic function. To verify this condition may take exponential time, but it gives quantitative expansivity results for random nets. Then the paper shows the universal approximation power for injective network and verify empirically that by enforcing a special structure that guarantees injectivity, they could improve the normal DCGAN training with better inception distance.

The paper has abundant theoretical results that cover different perspectives on understanding the role of injectivity in generative models.


1. The research problem on when a multilayer ReLU network is injective is mathematically interesting and sound. The paper has some though understanding of a single-layer case followed by some discussions of multi-layered settings. One main result of the paper is some sufficient and necessary condition of whether a single-layer ReLU network is injective, together with a lower bound for random net.
The paper further shows some sufficient property that grants an injective multi-layer generative model with a reasonable architecture.


However, I also have some concerns about the paper.

2. The considered problem seems not very useful to the ML community. It is not exactly true that global injectivity is important to the studies of inverse problems with generative models. The success of CSGM is from the fact that natural images lie in a small dimensional latent space. With leaky ReLU or flow model, global injectivity is automatically satisfied for non-degenerate weight matrices, and in most applications, we don't see much difference. With the ReLU network, I agree it was not clear whether a naturally trained generative model is injective or not, but the paper has not given a formative answer either. It is unclear to me how to interpret the results and how to make them useful in general.


3. The empirical study is quite interesting. However, it doesn't necessarily show that enforcing injectivity improves the quality of the generative model unless you could prove that the original DCGAN generates a network that doesn't satisfy injectivity. (Also, if you believe naturally trained generative model doesn't satisfy injectivity property, it again means that injectivity is not important to inverse problems since it works anyway and overruled the overall intuition of the paper). Also, the generated images w/o enforcing the structure doesn't seem to have any visual difference to me. Plus, the paper did not verify if the newly generated network has improved its performance on compressed sensing. If so, it could be an enlightening result and an interesting direction to explore.

---

> ### Author Response · Authors · 2020-11-18
> **To reviewer 4, benefits of global injectivity**
>
> *"The considered problem seems not very useful to the ML community. It is not exactly true that global injectivity is important to the studies of inverse problems with generative models. The success of CSGM is from the fact that natural images lie in a small dimensional latent space. With leaky ReLU or flow model, global injectivity is automatically satisfied for non-degenerate weight matrices, and in most applications, we don't see much difference. With the ReLU network, I agree it was not clear whether a naturally trained generative model is injective or not, but the paper has not given a formative answer either."*
>
> About your comment on global injectivity vs small-dimensional latent space—indeed, we agree—our results show injectivity precisely on this small-dimensional latent space (for example a low-dimensional manifold), rather than the full ambient space of images (which would not be useful and would be hardly possible). We state this in the revised manuscript at the beginning of Section 1.2. This is exactly parallel to injectivity for sparse vectors in compressed sensing. But there are actually multiple other ways in which injectivity is important for solving inverse problems with neural networks. For example, in CSGM, if the generator is injective we get that its range is a “true” manifold rather than some other low-dimensional structure. This immediately gives _global_ uniqueness and stability guarantees for compressed sensing (with our results on bi-Lipschitz constants) via existing results on random projections of manifolds (e.g. Baraniuk, Wakin). It is not clear how to get such guarantees otherwise. Another example is CSGM algorithms which use generator inversion as a step (as noted by Lei et al.; if linear programming inversions are used then in fact layerwise injectivity becomes important). Another example is simple direct supervised learning of the inverse map, without generative models. Suppose that a forward operator $A$ (e.g. a compressed sensing matrix) maps images from $R^n$ to compressive measurements in $R^m$ (again assuming that the images lie on some low-dimensional structure in $R^n$). Then since A is a Lipschitz map, its inverse must be injective! This may sound like a trivial statement but it shows that even in this direct inversion approach injectivity plays a role (we added a comment to that effect at the end of Section 1.2). Provable injectivity therefore allows practitioners in the ML community working in critical applications like medical imaging to avoid non-injective designs where two different reconstructions of say a tumor are equally plausible. Further, in terms of usefulness to the ML community, we feel we make a number of relevant contributions. For example, our universal approximation result directly shows that arbitrary manifolds can provably be fitted with networks whose range is itself a manifold (injective networks); thus we can also parameterize densities on such manifolds, or study topology of data. Our results on random matrices fold into the budding literature on neural networks with random weights (for example https://papers.nips.cc/paper/2017/file/0f3d014eead934bbdbacb62a01dc4831-Paper.pdf). These last are theoretical papers which shed light on the internal mechanics of neural networks. While they may not immediately propose algorithms which beat the state-of-the-art in some application, they further our understanding of neural networks which in turn leads to better algorithms and better empirical work.
>
> While it is true that one-to-one activation functions only require non-degenerate weight matrices, ReLUs are ubiquitous in practice. We believe that characterizing the properties of networks that are widely used in practice is important. As for giving a formative answer, we emphasize that our Corollary 2 provides a direct way of constructing provably (and optimally expansive) injective layers and in fact, we use this in our numerical experiments on inference. We added a discussion to this effect after Corollary 2. New experiments in Appendix A.3 similarly show issues with networks that were previously thought to be invertible.

---

> ### Author Response · Authors · 2020-11-18
> **To reviewer 4, Generative models and Injectivity**
>
> [please note that due to the way OpenReview posts comments, this is the second part of the response; the first part is below]
>
> *"it doesn't necessarily show that enforcing injectivity improves the quality of the generative model unless you could prove that the original DCGAN generates a network that doesn't satisfy injectivity."*
>
> As stated in the beginning of our numerical section, our aim is not to improve the quality of the generative model (we actually show that it remains the same) but to show that due to provable injectivity our inference performance is significantly better. Inferring latent codes from generated images of a GAN is an important problem and can only be solved if the generative model is injective. You are right that the experiment would be stronger if we proved that some of the DCGAN layers are not injective. Alas, per discussion at the end of 2.1 we do not have an efficient algorithm for this.
>
> *"prove that the original DCGAN generates a network that doesn't satisfy injectivity."*
>
> The Conv 1, 2 and 3 layers (Figure 1, Radford et al) in the original DCGAN paper use ReLU activations in their generator architecture. Note that the dimension after each of the conv 1, 2, and 3 layers doubles. By Cor 2, a ReLU layer with an expansivity of $2$ is injective only if it has the structure prescribed in Cor 2. Therefore, the DCGAN convolution layers are provably non-injective.
>
> *"Also, if you believe naturally trained generative model doesn't satisfy injectivity property, it again means that injectivity is not important to inverse problems since it works anyway and overruled the overall intuition of the paper."*
>
> As we mentioned in our introductory response, we can draw a parallel with “classical” compressed sensing. CS is used all the time with deterministic matrices for which no strong theoretical guarantees can be given (such as via spark or the restricted isometry property). This however does not diminish the importance of the seminal theoretical results in the 2000s which give reassurance that the method has a solid foundation, and a framework to think about designing “good” CS matrices. It is likely that a number of deterministic designs do yield non-unique reconstructions for well-chosen vectors. It then depends on the sensitivity of the application whether this is considered a serious problem or not. In fact, as we show in our new experiments in Appendix A.3, our theory gives us a tool to target pathologies caused by a lack of injectivity. Such pathologies could be disastrous in critical applications like medical imaging.

---

### Official Review · AnonReviewer2 · 2020-10-29

**Rating:** 5
**Confidence:** 2

**Review:**

Summary

This paper studies injectivity of fully-connected and convolutional ReLU networks. The paper studies conditions under which each layer of the ReLU network becomes injective for all possible inputs (hence making the entire network injective). Achieving injectivity requires that the output dimension of the layer is wider than input dimension at least by an expansion factor of 2 (Theorem 1, Corollary 2). Moving on to the injectivity of the entire network, the paper also shows that injective neural networks can universally approximate any continuous function $f : \mathbb R^n \to \mathbb R^m$ on a compact set $\mathcal Z \subset \mathbb R^n$ if $m \geq 2n+1$ (Theorem 3). Also, by applying random projections between hidden layers to reduce dimensions without losing injectivity, the paper shows that global injectivity of the network can be achieved (almost surely) with an input-output expansion factor of $\approx 2$ (Corollary 3).

Overall Assessment

I’m completely new to this topic of injective neural networks, and I’m not so familiar with application areas such as compressed sensing or inverse problems mentioned in this paper. So please take my words with a grain of salt.

It looks to me that the paper investigates a topic of interest, but at the same time, I question if this investigation can yield any fruitful outcome in practice. Also, I think the theory part of the paper has room for improvement in terms of clarity. For these reasons, I’m slightly leaning towards rejection at the moment, but I’m happy to raise my score after discussions with authors and other reviewers.

Detailed Comments

The paper puts extensive efforts to characterize injectivity on ReLU networks. Preliminary experiments look promising, but a more thorough experimental investigation in the application areas would have been a great addition to the paper.

Given all the theoretical results I have some fundamental questions on how to apply these ideas to practice. First of all, the paper analyzes the condition for ReLU layers to be injective. However, given that the DSS condition takes exponential time to check, how do you check for injectivity of a given network?

Also, a good portion of the paper deals with randomly sampled or generic matrices, but after you train a network using some training dataset, these matrices are no longer random or generic. How do you ensure that the network is still injective? Does the paper provide any method that makes sure that the network stays injective even after training? I see that the experiments in the paper used a convnet counterpart of Corollary 2, but Corollary 2 only holds if the expansion factor is exactly two.

In addition, I think the paper has several typos in notation as well as unclear points in theorem statements. It’d be great to fix/clarify the following issues:

- The statements around Theorem 4 are not clear enough to me. The term “shift” in Eq (8) is not defined precisely. In Theorem 4, does “for any $P$” mean “for all $P$” or “there exists any $P$”? How do you define DSS for multi-indices in $\mathbb R^P$? I didn’t get what Theorem 4 is trying to deliver.

- A dumb clarification question: it looks to me that Theorem 5 holds for any depth $L \geq 1$. So if $L=1$, a model $W_2\phi (W_1 x)$ (omitting bias terms) will represent $(x, F_{\theta} (x))$, where $F_{\theta}(x)$ approximates the target function $f(x)$. But after that, the construction requires $n$ projections $P_1, \dots, P_n$ from $\mathbb R^{n+m}$ to $\mathbb R^{m}$, which may sound like “multiple layers.” Are you viewing these projections as a single layer because the matrices $W_2, P_1, \dots, P_n$ can be multiplied all together?

- In the Notation paragraph in Section 1.3, the vector $m = (m_1, \dots, m_{L-1})$ is never used in the definition. Also, Eq (2) has $W_{L+1}$ in its definition which should be of dimension $\mathbb R^{n_{L+2} \times n_{L+1}}$. But two lines below that we have $n_L = m$, suggesting that the indices are off by one or two. Also, Eq (2) has a bias term $b_\ell$ which should be a typo.

- Corollary 1 states $\phi_\ell(W_{\ell} \cdot + b_\ell) : \mathbb R^{n_{i+1}} \to \mathbb R^{n_i}$ which should be corrected to $\mathbb R^{n_i} \to \mathbb R^{n_{i+1}}$.

- In the beginning of Section 2.3, $\| {\rm ReLU}(x) - {\rm ReLU}(y) \|$ should be corrected to $\| {\rm ReLU}(Wx) - {\rm ReLU}(Wy) \|$.

- In Section 2.4, the notation $I$ for a multi-index overloads with the identity matrix $I$.

---

> ### Author Response · Authors · 2020-11-18
> **To Reviewer 2, Usefulness, Clarity and Experiments**
>
> *"I question if this investigation can yield any fruitful outcome in practice."*
>
> The question of injectivity is central to the uniqueness of solutions to inverse problems which include critical applications like medical imaging. While our work is primarily a theoretical investigation into the problem, Corollary 2 provides the first necessary and sufficient condition on designing injective ReLU layers. Our Theorem 5 shows that the designed injective ReLU networks remain universal approximators which is a fundamental result that allows practitioners to not worry about expressivity of such networks. This further has immediate consequences for modeling distributions on manifolds. In fact, our numerical experiments, which use Corollary 2, show that adding injectivity constraints to the network gives equivalent performance while improving the inference of GANs. We further added Appendix A.3 which numerically shows the pathologies of inverting a network that is not globally injective. The definition of the DSS condition allows us to directly pinpoint such pathologies, which we feel is relevant in applications. This can perhaps be compared to adversarial examples, except that our experiment shows that entire regions of images cannot be reconstructed from the range of a network which fails to satisfy the DSS condition. Thus, although we believe that theoretical work which analyzes properties deep neural networks in important applications has merit on its own account, we also believe that the above examples show direct practical benefits of our results.
>
>
> *"I think the theory part of the paper has room for improvement in terms of clarity."*
>
> In order to improve clarity, we made a number of improvements to the formal statements, notation, motivation of Definition 1, and the implications of Corollary 2, Theorem 4, Theorem 5, and Corollary 3.
>
>
> *"Preliminary experiments look promising, but a more thorough experimental investigation in the application areas would have been a great addition to the paper."*
>
> We do agree that additional experiments would be welcome. Our main focus here is to provably characterize injectivity of ReLU networks. The paper is therefore dense with new theory regarding per-layer injectivity for fully-connected and convolutional layers, universal approximation results and connections to random matrix theory. Our previous experiment on inference does concern an important application area with generative models---this is not straightforward as GANs are not injective by design. We show that a simple change (guided by theory) leads us to better inference performance with no change in optimization or training hyperparameters. In the revised manuscript, we added further experiments which showcase the issues of inverting layers that are not injective (but were previously thought to be). In applications like medical imaging, absence of provably unique reconstructions, as shown in A.3, could be a serious problem.

---

> ### Author Response · Authors · 2020-11-18
> **To reviewer 2, Practical Questions, Training, and Typos.**
>
> [please note that due to the way OpenReview posts comments, this is the second part of the response; the first part is below]
>
> *"Given all the theoretical results I have some fundamental questions on how to apply these ideas to practice. First of all, the paper analyzes the condition for ReLU layers to be injective. However, given that the DSS condition takes exponential time to check, how do you check for injectivity of a given network?"*
>
> While DSS could take exponential time to check (per our discussion in the last paragraph of Section 2.1, although there is no proof as of yet), it is valuable as a design tool as well as a framework to think about injectivity. An example of it being used as a design tool is Corollary 2 and the subsequent paragraph, which give minimally expansive matrices for which the DSS condition is built-in and guaranteed. Note that the problem of minimal expansion was attempted in previous works (Lei et al 2019), but Corollary 2 is the first to give an optimal result (a necessary and sufficient condition). (Nota that in the revised manuscript we expanded the discussion after Corollary 2 to show how it can be used to construct injective layers with any expansivity larger than 2.)
>
>
> *"...after you train a network using some training dataset, these matrices are no longer random or generic."*
>
> You are right. First let us say that most of our results do not rely on random matrices (Thm 1, 3, 4, 5 and Cor 2) unlike some of the previous work. In our random matrix results in Thm 2 we do not use random matrices to model a network at initialization. We rather consider them as models of generic network layers. This is a common surrogate strategy / model in the literature since analyzing spectral properties of trained matrices is much more challenging. For example, Hand et al. 2017 and Lei et al 2019 analyze random-weight layers to gain insight into the behavior of generic networks. This is similar to compressed sensing which is often used with deterministic design matrices, but results on random matrices give us a framework to analyze and reason about expected performance.
>
> To complement these arguments, let us mention that analyzing neural networks with random weights is an important recent research direction, which aims to provide insights into spectral properties of neural networks by connecting them to random matrix theory. Excellent examples are the work on nonlinear random matrix theory for deep learning from Google Brain (Pennington et Worah 2017) or the work of Benigni et Péché 2019 on the eigenvalue distribution of certain nonlinear models of random matrices. This is especially important in the context of the recent advances on Neural Tangent Kernel [Jacot et al 2018] or lazy training [Chizat-Bach 2019] where, for a suitably initialized neural network, the weights change very little during training. In these cases our random matrix results in fact do shed light on the trained networks as well (although this is not our primary intention). We now make these connections in the manuscript just after the statement of Theorem 2.
>
>
> *"...paper has several typos in notation as well as unclear points in theorem statements. It’d be great to fix/clarify the following issues"*
>
> Convolution notation: To make the notion of a shift for the convolution result more clear, we added a new figure to illustrate the result and the various quantities in two dimensions.
>
> Collapsing projection layers into one: You are absolutely right. These matrices can all be multiplied together and thus correspond to a single linear layer. We added a corresponding clarification around Theorem 5. The reason to write it as multiple projections is that we wanted to hint at the proof technique which finds these projections sequentially.
>
> Finally, many thanks for the very detailed reading and discovery of issues with notation and misprints. These are all corrected in the updated manuscript.

---

### Official Review · AnonReviewer3 · 2020-10-30
**Comprehensive study of injective ReLU networks**

**Rating:** 8
**Confidence:** 3

**Review:**

This paper studies injective ReLU networks, motivated by various applications such as generative modeling, inverse problems and compressed sensing with generative priors. The authors fully characterize injectivity of fully-connected and convolutional layers and networks. They provide layerwise and multilayer results, characterizing the stability of inverting an injective network, and using tools from differential topology to study injectivity of deep networks. They also prove a sufficient condition for injectivity, which is an end-to-end doubling of the dimension.

The overall writing is clear and the use of pictorial illustration is helpful for less mathematically mature readers. Despite not being an expert in this area, I acknowledge the signifcance of exploring the properties and theories of injective (ReLU) neural networks, particularly for the study of inverse problems and generative modeling. The authors performed a comprehensive study of this subject in this paper, developing theories which allow deeper understanding of injective ReLU networks and their applications to nonlinear inverse and inference problems.


Pros:
- This paper established a mathematically rigorous framework to study the injectivity of fully-connected and convolutional ReLU networks.
- Substantial and careful discussion of common operations in fully-connected and convolutional neural networks, such as normalization strategies and pooling operations, regarding their effects on the injectivity of networks with their presence.


Cons:
- Would like to see an experiment for inverse problem applications, though this is rather minor.


Typos:
- In **Notation.**, do you mean $ \mathcal{NN}(n, m, L, \mathbf{n}) $ instead of $ \mathcal{NN}(n, m, L, \mathbf{m}) $?
- In (2), $ b_L $ instead of $ b_\ell $? Should there be $ b_{L+1} $ as well?
- Should the constant $ C(W) $ appear in (6)?
- Page 6, line 3: do you mean $ c \in \mathbb{R}^{N_1} \times \cdots \times \mathbb{R}^{N_p} $ instead?

---

> ### Author Response · Authors · 2020-11-18
> **To reviewer 3**
>
> We thank the reviewer for a positive evaluation. We would like to mention that we added a denoising experiment with one layer to showcase issues of non-injectivity for inverse problems. Finally, thank you for pointing out the misprints, all of which we corrected in the revised manuscript.

---

### Official Review · AnonReviewer5 · 2020-11-03
**The proposed directed spanning set (DSS) for injective ReLU networks is very useful. However, its advantage over prior work (e.g., Lei et al., 2019) is not well tested.**

**Rating:** 5
**Confidence:** 3

**Review:**

Summary:
Inverting a deep generative model with the ReLU activation function is very important. The paper studies injectivity of neural networks, which improves inference in GANs. Notably, the paper presents the directed spanning set (DSS), which is the core of all theoretical results. Some conditions for layerwise injectivity are also presented. Experiments verify that injective GANs improve inference.

Strengths:
+ Injectivity produces invertibility for ReLU networks. The paper defines the directed spanning set (DSS) to construct layerwise injectivity, which is very useful.
+ Some conditions for injective networks are presented.

Weaknesses:
- The paper is not clearly written, making it is hard to follow. There are two main reasons. First, the motivation of directed spanning set (DSS) is not clear. When Wx=y has multiple solutions or Wx<0 has solutions, ReLU(Wx) will precludes injectivity, where y>=0 is a constant. Are the two cases related to DSS? Second, the notations are confusing. For example, W is a matrix and is also a set. w is a row vector, while z and b are column vectors. I is an identity matrix and is also an inference network.
- The main theoretical results (i.e., Corollary 2 and Theorem 2) are highly related to (Lei et al., 2019). However, the paper did not compare with (Lei et al., 2019). What is the advantage of the proposed results over those in (Lei et al., 2019)? It seems that the result in (Lei et al., 2019) (i.e., m>=2.1n) is tighter than the presented one (m>=5.7n).
- Injectivity for Gaussian weights should be tested by comparing with gradient descent and (Lei et al., 2019). Also, how to use Corollary 2 and Theorem 2 to construct W in practice?
- It is better to test with some real problems, e.g., denoising.

Minor comments:
1.	In the first paragraph of subsection 2.3, ||ReLU(x)-ReLU(y)|| should be ||ReLU(Wx)-ReLU(Wy)||.
2.	What do C(W) and S(x,W) mean in Theorem 3?

---

> ### Author Response · Authors · 2020-11-18
> **To reviewer 5**
>
> *"the motivation of directed spanning set (DSS) is not clear. When Wx=y has multiple solutions or Wx<0 has solutions, ReLU(Wx) will precludes injectivity, where y>=0 is a constant. Are the two cases related to DSS?"*
>
> We are not sure that we fully understand your concern about DSS---if the explanation offered below does not address it, please let us know. Indeed, the examples you mention are captured within the definition of DSS. However, non-singularity of the W matrix is not sufficient, hence we need the full combinatorial statement used in the DSS definition. Satisfying the DSS property at a point is equivalent to being injective w.r.t. that point, as indicated by Theorem 1. To make this point clearer in the manuscript, we added additional context: the first paragraph of Section 2.1 is expanded to show that the question of injectivity of a ReLU layer is subtle, using both of your suggested examples in addition to Figure 2.
>
> *"notations are confusing."*
>
> Following your comment we removed the overloaded set notation in order to make the paper easier to read. We only keep the notation $w \in W$ to mean that w is a row vector of W as it makes a number of statements much less cumbersome. We further scrutinized the manuscript for confusing notation: we now use different symbols for the index $I$, the identity matrix $Id$ and an inference network $I_{net}$. Thank you for pointing out these clashes.
>
> *"However, the paper did not compare with (Lei et al., 2019)...is tighter than the presented one (m>=5.7n)"*
>
> We do mention in the paragraph before Theorem 2 that the proof of Lei et al. shows that a ReLU layer is injective about a single fixed point in its range with high probability provided that $c \geq 2.1$. More concretely, they show that given a ReLU layer with an $m \times n$ iid Gaussian matrix, a fixed point in $R^n$ will have at least n positive inner products with its rows when $m > 2.1n$ (with high probability). This, however, does not guarantee injectivity (that is to say, invertibility on the range), which in turn requires that for any two distinct inputs the layer produces distinct outputs. To appreciate the relevance of this, note that, for example, a successfully trained GAN maps an iid Gaussian in the latent space to the target distribution in data space. Our generated data is expected to come from any input latent code, not just one particular one. In order to make these comparisons stand out more prominently we added more numerical experiments (Appendix A.3). These show that with the expansivity of 2.1 there are many points in the range about which the network is not invertible. Finally, let us mention again that our Corollary 2 gives a completely deterministic result with better expansivity than any random Gaussian layer, namely exactly 2.
>
> *Minor comments*
> We addressed all of your minor comments in place; thank you for helping us improve the manuscript.

---

### Author Response · Authors · 2020-11-18
**Summary for all reviewers**

We would like to thank the referees for taking the time to read and comment on our paper. Here we summarize our comments and the revisions to the manuscript. Detailed responses are given to each reviewer’s comments individually.

*Motivation for studying injectivity of neural networks for inference and inverse problems*

We wish to emphasize that we are mainly concerned with the theoretical characterization of injectivity and invertibility of neural networks. As we argue in the paper and in the individual responses below, this is instrumental to understanding when application of neural networks to inverse and inference problems is well-posed and to provide performance guarantees. While understanding our results is not per se necessary to do compressed sensing with generative models, knowing that there exist guarantees that this procedure succeeds in certain settings is important for both theoreticians and practitioners, especially in sensitive applications such as medical imaging. We draw a parallel with the compressed sensing literature: l1 minimization was known to "just work" to recover sparse much before the major theoretical advances in the 2000s. These theoretical results, however, were important both as a vehicle to generate further practical advances and as a “certificate” that the method rests on firm foundations.


*Additional numerical results*

The main goal of our numerical experiments is to illustrate the properties and benefits of injective networks rather than to beat state-of-the-art models in compressed sensing applications. We stress that our theory is not limited to CS applications but is more fundamental—when is any multi-layer ReLU network injective? What can injective networks approximate? How stable is the inversion? How to construct convolutional injective layers? Addressing the comments of several reviewers, in the revised manuscript we have added further numerical experiments which show that previous theoretical results about the 2.1 expansivity are not sufficient to invert the layer by constructing counterexamples. Even so, we would like to state that we again see the new experiments as a complement of the theory and as a way to illustrate it. In particular, we show that DSS leads to injectivity and that inversion becomes unstable in its absence.


*Summary of major changes in the manuscript*

Added Appendix A.3 to show that the expansivity factor of 2.1 from previous work is not enough to invert a ReLU layer.
Changed and simplified notation as per the suggestions of reviewers. We are very grateful for the very useful suggestions and comments.

---

> ### Author Response · Authors · 2020-11-24
> **Fixed a typo in the proof of Theorem 5**
>
> Dear reviewers, this is a note that we fixed a small typo regarding the bound on sums of binomial coefficients used in the proof of Theorem 5 and made a corresponding modification in the statement.

---

### Decision · Program_Chairs · 2021-01-07
**Final Decision**

**Decision:**

Reject

**Comment:**

The average review ratings for this paper is somewhat borderline. The paper provides mathematical characterizations on when ReLU neural networks are injective. The paper has very nice ideas, but the reviewers also pointed out several key concerns:

1. “Given that the DSS condition takes exponential time to check, how do you check for injectivity of a given network?”
2. “But after you train a network using some training dataset, these matrices are no longer random or generic. How do you ensure that the network is still injective?”
3. “With leaky ReLU or flow model, global injectivity is automatically satisfied for non-degenerate weight matrices, and in most applications, we don't see much difference”

I think points 2&3 are particularly important here. It seems that for practitioners, if injectivity is a key concern, then one can just use leaky-relu with well-conditioned weight matrices that guarantee injectivity. Note that well-conditioning is easy to check and relatively easy to enforce. It’s unclear to the AC why one has to stick to particularly the relu activation and the recipe provided by corollary 2 and the paragraphs below Corollary 2. (It also seems to the AC that Corollary 2’s construction is fundamentally similar to the using leaky-relu, but the AC is not quite sure.) Given that a much easier workaround (using leaky-relu and full-rank matrices) is available and is widely used in prior works (when it’s necessary), the AC, unfortunately, does not see that the paper could have a strong impact on the ML community and does not think the experimental results are sufficient to justify that this is a better idea than using leaky-relu. In the AC’s opinion, the paper might be more compelling in a math venue.